# Effects of Different Water Table Depth and Salinity Levels on Quality Traits of Bread Wheat

İsmail Sezer [1], Hasan Akay [1,*], Zeki Mut [2], Hakan Arslan [3], Elif Öztürk [1], Özge Doğanay Erbaş Köse [2] and Mehmet Sait Kiremit [3]

1 Department of Field Crops, Faculty of Agriculture, Ondokuz Mayıs University, Samsun 55270, Turkey; isezer@omu.edu.tr (İ.S.); elif.ozturk@omu.edu.tr (E.Ö.)
2 Department of Field Crops, Faculty of Agriculture and Natural Sciences, Bilecik Şeyh Edebali University, Bilecik 11230, Turkey; zeki.mut@bilecik.edu.tr (Z.M.); ozgedoganay.erbas@bilecik.edu.tr (Ö.D.E.K.)
3 Department of Agricultural Structures and Irrigation, Faculty of Agriculture, Ondokuz Mayıs University, Samsun 55270, Turkey; hakan.arslan@omu.edu.tr (H.A.); mehmet.kiremit@omu.edu.tr (M.S.K.)
* Correspondence: hasan.akay@omu.edu.tr

**Abstract:** Abiotic stress factors encountered in production lands influence both the yield and the quality traits of bread wheat. This study investigated the effects of three different water table depths (30, 55, and 80 cm) and four different groundwater salinity levels (0.38, 2.0, 4.0, and 8.0 dSm$^{-1}$) on some quality traits of bread wheat under irrigated and unirrigated conditions. The experiments were conducted in the 2018 and 2019 growing seasons in randomized blocks—factorial (three factors) experimental design with three replications under controlled conditions. The hectoliter weight, fat ratio, starch ratio, protein content, Zeleny sedimentation, wet gluten content, ash ratio, acid detergent fiber (ADF), and neutral detergent fiber (NDF) values were investigated. The hectoliter weights varied between 66.1 and 77.8 kg, fat ratios between 1.49% and 1.70%, starch ratios between 61.9% and 67.8%, protein contents between 11.9% and 13.8%, Zeleny sedimentation values between 23.5 and 28.0 mL, wet gluten contents between 25.0% and 28.8%, ash ratios between 1.43% and 1.75%, and ADF values between 2.85% and 4.12%. The quality traits were positively influenced by increasing the water table depths. With increasing the groundwater salinity levels, the hectoliter weight, fat ratio, starch ratio, and NDF values decreased, while the protein ratio, sedimentation value, wet gluten content, ash ratio, and ADF values increased.

**Keywords:** bread wheat; water table; salinity; gluten; sedimentation





## 1. Introduction

Wheat is among the most widely cultivated agricultural crop worldwide. It constitutes the primary calorie source in human nutrition [1,2]. Annually, 766 million tons of wheat are produced every year globally, and 19 million tons of wheat are produced in Turkey [3]. Rain-fed farming is practiced in the wheat cultivation of arid and semi-arid regions, and the yields are decreasing significantly because of insufficient water resources [4].

In the Mediterranean climate zone, producers generally practice one or two supplementary irrigations in a year (except for dry years) in wheat fields using the surface irrigation method. In these regions, the March and May months coincide with the flowering and milk dough stages of wheat, which are the sensitive growth periods. Insufficient precipitations in these months may result in serious yield losses [5]. The initiation of irrigations in arid and semi-arid regions subsequently brought about drainage problems, and such problems then resulted in the rise of the water table and salinity problems [6]. Global warming is the most challenging environmental problem that humanity should deal with. Global warming alters the seasonal normal and increases soil salinity through insufficient precipitations and high evaporations. The water table and salinity control in these regions are largely dependent on a well arrangement of the water balance.

Salinity problems could be overcome with a well water balance [7]. Salinity is among the most significant problems encountered in agricultural fields worldwide. Salinity-induced yield decreases are experienced in various parts of the world, and salinity ultimately terminates agricultural practices. High irrigation water salinity or soil salinity raises the osmotic pressure of the soil solution, then reduces the water uptake of roots from the soil and, consequently, decreases the crop yield and quality [8]. Therefore, crop and soil-based water management strategies should be developed to sustain water and soil resources.

The wheat yield was decreased by approximately 17% with increasing the irrigation water salinity from 0.6 to 10 dS/m [9]. Besides the protein content, the wet and dry gluten content values of the wheat crops were increased with the increasing salinity and drought stresses [10]. Extreme droughts can cause significant decreases in the protein content, wet gluten content, and sedimentation volume content of wheat [11].

Several factors designate the wheat quality, and the quality criteria vary significantly based on the producer, industry, and consumer demands [12,13]. The protein ratio is the most important quality criterion in wheat [14], and the protein ratios of different wheat varieties under different environmental conditions vary between 6 and 22% [15]. In bread wheat, the sedimentation value and wet gluten ratio designate the protein quality. Therefore, besides the protein ratio, the protein quality also plays a great role in the quality of bread wheat [16]. The protein ratio and quality may change with the growing conditions and climate factors [17]. In Turkey, the hectoliter weight of wheat varies between 70 and 84 kg and is mainly dependent on the cultivars and climate conditions. The starch content of wheat grain constitutes about 65–70% of the grain dry weight [13]. Acid detergent fiber (ADF) is an indicator of the cellulose, lignin, and insoluble protein contents of the cell wall and reveals information about the digestibility of the products [18]. Neutral detergent fiber (NDF) expresses the indigestible substances like cellulose, hemicellulose, lignin, cutine, and insoluble protein of the cell wall. High NDF values negatively influence the feed quality [19].

Wheat production, which is one of the most basic foods globally, is severely affected by drainage and salinity problems. Moreover, the quality parameters are as important as the yield in wheat.

In the literature, many studies have been carried out on the effects of the water table depth and salinity on the quality parameters of wheat. However, there is no detailed study on the combined effects of these stress factors on the quality parameters of wheat. Therefore, the present study was conducted to investigate the effects of different water table depths and salinity levels on the quality parameters of bread wheat plants under with and without irrigation conditions.

## 2. Materials and Methods

### 2.1. Experimental Site Description

This study was conducted at the Agricultural Research and Implementation Center (41°21′ N and 36°11′ E, elevation 192 m above sea level), which belongs to the Faculty of Agricultural, University of Ondokuz Mayıs, Northern Turkey. The experiment was laid out over a two-year growing season, from December to June 2018 to 2019. The lysimeters were conducted on a four-sides open land area (120 m$^2$) with a plastic cover above to protect from precipitation. The daily temperature and relative humidity values were measured with a datalogger located at the middle of the research area from 2 m above the ground for two growing seasons. The mean monthly temperature and relative humidity data are illustrated in Table 1.

**Table 1.** The average monthly temperature and relative humidity values in the first and second growing seasons.

|  | November | December | October | February | March | April | May | June |
|---|---|---|---|---|---|---|---|---|
| **Min Temperature (°C)** | | | | | | | | |
| 2017 to 2018 | 3.4 | 2.1 | 0.5 | 2.5 | 0 | 4.6 | 7.7 | 12.3 |
| 2018 to 2019 | 4.5 | 0.8 | −0.2 | 0.6 | 0.2 | 3.2 | 8.1 | 15.8 |
| **Average Temperature (°C)** | | | | | | | | |
| 2017 to 2018 | 9.7 | 11.8 | 7.7 | 9.3 | 11.6 | 16.6 | 19.2 | 24.4 |
| 2018 to 2019 | 11.6 | 8.8 | 8.1 | 8.1 | 8.9 | 12.7 | 18.2 | 23.9 |
| **Max. Temperature (°C)** | | | | | | | | |
| 2017 to 2018 | 16.3 | 22.3 | 23.6 | 25.4 | 28.4 | 28.8 | 34.9 | 38.4 |
| 2018 to 2019 | 23.7 | 22.3 | 21 | 22 | 25.9 | 32.6 | 30.6 | 33.7 |
| **Average Relative Humidity (%)** | | | | | | | | |
| 2017 to 2018 | 73.1 | 64.8 | 76 | 80 | 77 | 68.2 | 79.7 | 66 |
| 2018 to 2019 | 77.5 | 78.2 | 70.9 | 79.8 | 74.3 | 78.6 | 81.7 | 83.8 |

The experimental soil was obtained from the top 30-cm layer, and its texture was loam with 25.3% clay, 31.3% silt, and 43.4% sand. Additionally, its chemical properties were 2.4% organic matter, 7.1-mg kg$^{-1}$ phosphorus, 0.33-me 100 g$^{-1}$ potassium, 7.99 pH, and 0.27-dSm$^{-1}$ electrical conductivity. Phosphorous (P) was determined with a UV–Visible spectrophotometer according to Reference [20]. Potassium was measured using flame photometers.

*2.2. Lysimeter Set-Up*

A 5-cm layer of gravel and a 5-cm layer of sand were placed at the bottom of each lysimeter to provide a continuous water supply from Mariotte bottles to lysimeters. Then, each lysimeter was filled with 330 kg of soil sieved through a 4-mm sieve, and the soil in the lysimeter was compacted layer by layer (10 cm) to reach a soil bulk density of 1.297 gr/cm$^3$. Schematic views of the lysimeters used in the present experiment are presented in Figure 1 [21]. The groundwater depths in the lysimeters were controlled at the constant levels of 30, 55, and 80 cm from below the soil surface. The groundwater was checked daily by keeping the water in the Mariotte bottles at a constant level. The daily amount of water moving into each lysimeter was calculated by water loss from the Mariotte bottle. The drainage pipe was placed above the groundwater depth into each lysimeter to drain out excess water automatically.

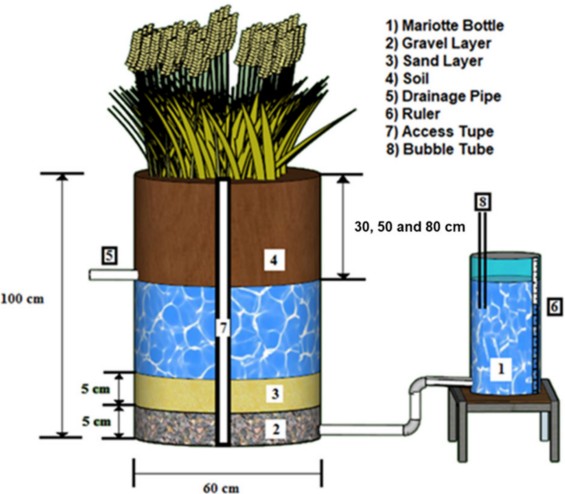

**Figure 1.** Schematic view of lysimeter and Mariotte bottle used in the study.

### 2.3. Experimental Design and Treatments

The experimental traits were conducted in 72 lysimeters, which were 100 cm deep with 60-cm inner diameters. The experimental design was an arrangement in a randomized complete block with an irrigation treatment as the main plot, groundwater depth as the subplot, and groundwater salinity as the sub-subplot with three replicates. The experimental traits contained two irrigation treatments of $I_1$ (with irrigation) and $I_2$ (without irrigation) and three groundwater depths of 30 cm, 55 cm, and 80 cm and four groundwater salinities of 0.38 $dSm^{-1}$, 2.0 $dSm^{-1}$, 4.0 $dSm^{-1}$, and 8.0 $dSm^{-1}$ (Table 2).

**Table 2.** Experimental treatments of the irrigation treatments, groundwater depth, and groundwater salinity.

| Irrigation Treatments | Groundwater Depth (cm) | Groundwater Salinity ($dSm^{-1}$) |
|---|---|---|
| $I_1$ (with irrigation) | $D_1$ = 30 cm | $S_1$ = 0.38 $dSm^{-1}$ |
| $I_2$ (without irrigation) | $D_2$ = 55 cm | $S_2$ = 2.0 $dSm^{-1}$ |
| | $D_3$ = 80 cm | $S_3$ = 4.0 $dSm^{-1}$ |
| | | $S_4$ = 8.0 $dSm^{-1}$ |

At the end of the tillering period, the lysimeters were saturated from the bottom with different water salinities of 0.38, 2.0, 4.0, and 8.0 $dSm^{-1}$ at up to 30, 55, and 80-cm groundwater depths for two weeks. The saline waters were prepared with the use of highly soluble $MgSO_4$ (99% purity), $CaCl_2$ (99% purity), and NaCl (99.5% purity) salts. The amount of salt to be added to prepare relevant salt concentrations (EC values) was calculated with the use of QBASIC software to achieve a sodium adsorption ratio (SAR) of <5 and a Ca/Mg ratio of 1:3.

In both growing years, all lysimeters were supplemented with 10-mm irrigation water until the end of the tillering period. The first irrigation was applied after establishing the constant groundwater depths. For this, each lysimeter's volumetric soil moisture content above the groundwater depth was measured with a neutron scattering method (CPN 503 Dr Hydro probe). The polyvinyl chloride (PVC) pipes used for the neutron meter measurements were placed in the middle of the lysimeters. Under $S_1$ (with irrigation) conditions, the soil moisture content of each lysimeter was measured, and irrigation water was applied to bring the available soil moisture to field capacity. However, under $I_2$ (without irrigation) conditions, the soil moisture content of each lysimeter was measured, but irrigation water was not applied from the end of the tillering period to harvesting.

After the first irrigation, the volumetric soil moisture content above the groundwater depth in all the lysimeters was monitored every seven days. When 50% of the available soil water above the groundwater depth was depleted, irrigation water was added to fill up to field capacity in the $I_1$ treatments.

### 2.4. Crop Management

A Pandas wheat cultivar was used as the seed material of the study. In the first year, wheat seeds were sown on 11 November 2017, and at 14 November 2018 in the second year, to have a sowing density of 500 seeds per $m^2$. Fertilization was applied according to the soil analysis. With this aim, 100-kg $ha^{-1}$ pure nitrogen (N) and 60-kg $ha^{-1}$ $P_2O_5$ were applied to each lysimeter. All phosphorus was applied in diammonium phosphate form prior to sowing. Additionality, nitrogen was applied in the form of urea (46% N) at two different times; half of the nitrogen was applied at sowing, and the other half was applied just before the bolting period. Weed control was practiced manually. Wheat crops from each lysimeter were harvested at full maturity on 8 June 2018 and 19 June 2019.

### 2.5. Grain Quality Parameters

The quality indicators included the hectoliter weight (kg), ash content (%), fat content (%), protein content (%), Zeleny sedimentation (mL), and wet gluten values (%). These parameters were determined following Reference [22]. The acid detergent fiber (ADF) (%)

and neutral detergent fiber (NDF) contents (%) were determined according to Reference [23] and the starch ratio was determined with the use of the Ewers Polarimetric method [24].

### 2.6. Data Analysis

Firstly, all data were subjected to the homogeneity test, and they were shown normal distribution. The experimental data were subjected to a variance analysis using JMP statistical software (SAS Institute, Cary, NC, USA) [25]. All treatment means were compared using Tukey's test. Biplot and Pearson correlation analyses were conducted to assess the relations among the investigated parameters.

## 3. Results

The variance analysis results of the effects of the irrigation level, groundwater depth, salinity, year, and their interactions on the quality parameters of wheat are given in Table 3.

**Table 3.** Variance analysis table of the analyzed parameters *.

| VS. | DF | Average of Squares | | | | | | | | |
|---|---|---|---|---|---|---|---|---|---|---|
| | | HW | SC | PC | ZSV | WG | FC | AC | ADF | NDF |
| Year | 1 | 534.15 ** | 179.23 ** | 292.75 ** | 4289.27 ** | 4112.02 ** | 0.38 ** | 0.043 ** | 1.41 ** | 10.79 ** |
| Irrigation | 1 | 147.42 ** | 0.09 | 0.20 | 22.18 ** | 6.18 ** | 0.01 | 0.054 ** | 0.84 ** | 0.05 |
| D | 2 | 222.73 ** | 14.51 ** | 6.36 ** | 39.40 ** | 18.40 ** | 0.16 ** | 0.071 ** | 1.57 ** | 3.86 ** |
| S | 3 | 196.38 ** | 34.05** | 7.18 ** | 32.29 ** | 22.41 ** | 0.07 * | 0.304 ** | 4.25 ** | 2.70 ** |
| Block | 4 | 0.09 | 0.27 | 0.14 | 0.04 | 0.08 | 0.03 | 0.002 | 0.01 | 0.02 |
| Y×I. Int. | 1 | 6.63 * | 4.87 ** | 0.23 | 6.17 ** | 0.10 | 0.09 * | 0.001 | 0.01 | 1.70 ** |
| Y×D. Int | 2 | 14.63 ** | 44.04 ** | 0.47 * | 0.61 ** | 0.30 * | 0.23 ** | 0.001 | 0.01 | 0.68 ** |
| Y×S. Int. | 3 | 0.09 | 1.39 ** | 0.30 * | 5.01 ** | 1.45 ** | 0.002 | 0.003 * | 0.03 ** | 0.18 ** |
| D×I. Int. | 2 | 14.29 ** | 8.77 ** | 1.13 ** | 13.53 ** | 0.07 | 0.04 | 0.002 | 0.18 ** | 0.27 ** |
| S×I Int. | 3 | 0.25 | 0.30 | 0.06 | 1.01 ** | 1.08 ** | 0.01 | 0.001 | 0.21 ** | 0.04 * |
| S×D Int. | 6 | 3.41 ** | 0.38 | 0.10 | 0.61 ** | 0.13 | 0.01 | 0.002 * | 0.02 * | 0.03 |
| Y×I×D Int. | 2 | 2.80 | 5.46 ** | 0.10 | 1.21 ** | 0.25 | 0.02 | 0.001 | 0.01 | 0.53 ** |
| Y×I×S Int. | 3 | 0.10 | 2.19 ** | 0.11 | 0.32 ** | 0.42 ** | 0.02 | 0.001 | 0.01 | 0.02 |
| Y×D×S Int. | 6 | 3.70 ** | 1.41 ** | 0.07 | 0.60 ** | 0.13 | 0.02 | 0.001 | 0.01 | 0.04 * |
| S×D×I nt. | 6 | 0.58 | 0.75 ** | 0.03 | 0.32 ** | 0.17 | 0.01 | 0.0002 | 0.04 ** | 0.15 ** |
| Y×I×D×S Int. | 6 | 0.85 | 0.82 ** | 0.10 | 0.14 ** | 0.09 | 0.01 | 0.0003 | 0.01 | 0.14 ** |
| Error | 92 | 0.99 | 0.19 | 0.11 | 0.07 | 0.10 | 0.02 | 0.0009 | 0.01 | 0.01 |
| % CV | | 1.37 | 6.81 | 2.58 | 1.03 | 1.19 | 8.73 | 1.87 | 2.07 | 6.50 |

* The $p < 0.05$ and ** $p < 0.01$ levels are significant. HW = Hectoliter weight (kg), FC: Fat Content (%), NC: Starch Content (%), PC: Protein Content (%), ZSV: Zeleny Sedimentation Value (mL), WG: Wet gluten (%), AC: Ash Content (%), ADF: Acid Detergent Insoluble Fiber (%), NDF: Neutral Detergent Insoluble Fiber (%), DF: Degree of Freedom, Y: Year, I: Irrigation, D: Depth of Ground Water, and S: Ground Water Salinity.

### 3.1. Hectoliter Weight

The hectoliter weights at different water table depths were significantly different ($p \leq 0.01$) (Table 3). The highest hectoliter weight (73.9 kg) was obtained from the $D_3$ level, followed, respectively, by the $D_2$ (73.3 kg) and $D_1$ (69.9 kg) levels (Table 4). Significant differences were also observed in the hectoliter weights at different groundwater salinity levels ($p \leq 0.01$). The greatest hectoliter weight (75.2 kg) was obtained from the $S_1$ level, followed, respectively, by the $S_2$ (73.1 kg), $S_3$ (71.5 kg), and $S_4$ (69.7 kg) levels. In terms of the irrigation treatments, the greatest hectoliter weight (73.4 kg) was obtained from the $I_1$ treatment and the lowest (71.4 kg) from the $I_2$ treatment. The differences in the hectoliter weights of the irrigation treatments were found to be significant ($p \leq 0.01$). In terms of the salinity x water table depth interactions (S × D), the greatest hectoliter weight (76.2 kg) was obtained from the $S_1 \times D_3$ combination and the lowest (66.9 kg) from the $S_4 \times D_1$ combination (Table 5). In terms of the D × I interactions, the highest hectoliter weight (75.6 kg) was obtained from the $D_3$ x $I_1$ combination and the lowest (69.3 kg) from the $D_1 \times I_2$ combination (Table 5). In the present study, the greatest hectoliter weight (77.8 kg) was obtained from the $S_1 \times D_3 \times I_1$ combination and the lowest (66.1 kg) from the $S_4 \times D_1 \times I_2$ combination (Table 6).

**Table 4.** The mean of the two years for the effects of the four salinity levels, three water table depths, and two irrigation levels on the wheat parameters *.

| Source of Variance | HW | SC | PC | ZSV | WG | FC | AC | ADF | NDF |
|---|---|---|---|---|---|---|---|---|---|
| | | | | Salinity (S) | | | | | |
| S$_1$ | 75.2 a | 65.1 a | 12.3 d | 24.8 d | 25.8 d | 1.64 a | 1.49 d | 3.04 d | 15.7 a |
| S$_2$ | 73.1 b | 64.3 b | 12.7 c | 25.4 c | 26.2 c | 1.62 a | 1.57 c | 3.33 c | 15.5 b |
| S$_3$ | 71.5 c | 63.6 c | 13.1 b | 26.3 b | 26.9 b | 1.63 a | 1.65 b | 3.59 b | 15.3 c |
| S$_4$ | 69.7 d | 62.8 d | 13.4 a | 26.9 a | 27.6 a | 1.59 b | 1.73 a | 3.85 a | 15.1 d |
| | | | | Water table depth (D) | | | | | |
| D$_1$ | 69.9 c | 63.3 b | 12.5 c | 25.2 c | 26.0 c | 1.56 B | 1.57 c | 3.27 c | 15.7 a |
| D$_2$ | 73.3 b | 64.3 a | 12.9 b | 25.5 b | 26.7 b | 1.67 A | 1.60 b | 3.45 b | 15.3 b |
| D$_3$ | 73.9 a | 64.3 a | 13.2 a | 26.9 a | 27.2 a | 1.63 A | 1.65 a | 3.63 a | 15.1 c |
| | | | | Irrigation (I) | | | | | |
| I$_1$ | 73.4 | 63.9 | 12.9 | 26.2 a | 26.8 a | 1.61 | 1.62 | 3.37 | 15.3 |
| I$_2$ | 71.4 | 64.0 | 12.8 | 25.4 b | 26.4 b | 1.62 | 1.58 | 3.53 | 15.4 |

* There is no difference in the significance level of 0.01 between the averages shown with the same letter in each column. HW = Hectoliter weight (kg), FC: Fat Content (%), SC: Starch Content (%), PC: Protein Content (%), ZSV: Zeleny Sedimentation Value (mL), WG: Wet gluten (%), AC: Ash Content (%), ADF: Acid Detergent Fiber (%), and NDF: Neutral Detergent Fiber (%).

**Table 5.** The mean of the two years for the interaction effects of the salinity level × water table depth, salinity × irrigation, and water table depth × irrigation on the wheat parameters *.

| Source of Variance | HW | PC | ZSV | WG | AC | ADF | NDF |
|---|---|---|---|---|---|---|---|
| | | Salinity (S) × Water Table Depth (D) | | | | | |
| S$_1$ × D$_1$ | 73.5 bc | 11.9 | 23.9 g | 25.2 | 1.45 g | 2.91 h | 16.0 |
| S$_2$ × D$_1$ | 70.1 fg | 12.3 | 24.5 f | 25.6 | 1.53 ef | 3.15 g | 15.8 |
| S$_3$ × D$_1$ | 69.1 g | 12.8 | 25.9 d | 26.3 | 1.63 d | 3.40 e | 15.6 |
| S$_4$ × D$_1$ | 66.9 h | 13.1 | 26.3 c | 26.8 | 1.66 cd | 3.63 d | 15.3 |
| S$_1$ × D$_2$ | 75.8 a | 12.4 | 24.5 f | 25.9 | 1.49 fg | 3.00 h | 15.6 |
| S$_2$ × D$_2$ | 74.3 bc | 12.8 | 25.2 e | 26.2 | 1.56 e | 3.30 f | 15.5 |
| S$_3$ × D$_2$ | 72.2 de | 13.1 | 25.8 d | 26.9 | 1.64 cd | 3.63 d | 15.3 |
| S$_4$ × D$_2$ | 70.8 f | 13.3 | 26.4 c | 27.7 | 1.71 ab | 3.87 b | 15.1 |
| S$_1$ × D$_3$ | 76.2 a | 12.8 | 25.8 d | 26.5 | 1.54 e | 3.22 fg | 15.5 |
| S$_2$ × D$_3$ | 74.9 ab | 13.1 | 26.5 c | 26.6 | 1.62 d | 3.55 d | 15.2 |
| S$_3$ × D$_3$ | 73.3 cd | 13.4 | 27.3 b | 27.5 | 1.68 bc | 3.74 c | 14.9 |
| S$_4$ × D$_3$ | 71.4 ef | 13.7 | 27.9 a | 28.3 | 1.73 a | 4.03 a | 14.7 |
| | | Salinity (S) × Irrigation (I) | | | | | |
| S$_1$ × I$_1$ | 76.2 | 12.3 | 24.9 e | 26.0 de | 1.51 | 3.03 e | 15.7 a |
| S$_2$ × I$_1$ | 74.0 | 12.8 | 25.8 d | 26.2 d | 1.59 | 3.31 d | 15.4 b |
| S$_3$ × I$_1$ | 72.6 | 13.2 | 26.8 b | 27.1 b | 1.67 | 3.49 bc | 15.3 c |
| S$_4$ × I$_1$ | 70.8 | 13.4 | 27.4 a | 28.1 a | 1.72 | 3.67 b | 15.0 d |
| S$_1$ × I$_2$ | 74.2 | 12.3 | 24.6 f | 25.7 e | 1.47 | 3.05 e | 15.7 a |
| S$_2$ × I$_2$ | 72.2 | 12.7 | 25.0 e | 26.1 d | 1.55 | 3.35 d | 15.5 b |
| S$_3$ × I$_2$ | 70.5 | 13.0 | 25.8 d | 26.7 c | 1.63 | 3.68 b | 15.2 c |
| S$_4$ × I$_2$ | 68.6 | 13.3 | 26.3 c | 27.1 b | 1.68 | 4.02 a | 15.1 d |
| | | Water Table Depth (D) × Irrigation (I) | | | | | |
| D$_1$ × I$_1$ | 70.5 d | 12.7 d | 25.7 d | 26.2 | 1.58 | 3.14 d | 15.7 a |
| D$_1$ × I$_2$ | 69.3 e | 12.4 e | 24.6 e | 25.8 | 1.55 | 3.40 c | 15.6 b |
| D$_2$ × I$_1$ | 74.1 b | 13.0 bc | 26.3 c | 26.9 | 1.62 | 3.36 c | 15.3 d |
| D$_2$ × I$_2$ | 72.4 c | 12.8 cd | 24.6 e | 26.4 | 1.57 | 3.53 b | 15.4 c |
| D$_3$ × I$_1$ | 75.6 a | 13.1 ab | 26.7 b | 27.4 | 1.67 | 3.62 a | 15.0 f |
| D$_3$ × I$_2$ | 72.3 c | 13.4 a | 27.0 a | 27.0 | 1.62 | 3.64 a | 15.2 e |

* There is no difference in the significance level of 0.01 between the averages shown with the same letter in each column. HW = Hectoliter weight (kg), PC: Protein Content (%), ZSV: Zeleny Sedimentation Value (mL), WG: Wet gluten (%), AC: Ash Content (%), ADF: Acid Detergent Insoluble Fiber (%), and NDF: Neutral Detergent Insoluble Fiber (%).

**Table 6.** The mean of the two years for the interaction effects of the salinity level × water table depth × irrigation on the wheat parameters *.

| Source of Variance | SC | ZSV | ADF | NDF |
|---|---|---|---|---|
| **Salinity (S) × Water Table Depth (D) × Irrigation (I)** | | | | |
| $S_1 \times D_1 \times I_1$ | 64.5 bc | 24.3 de | 2.85 f | 16.2 a |
| $S_2 \times D_1 \times I_1$ | 63.8 c | 25.0 d | 3.13 de | 15.9 ab |
| $S_3 \times D_1 \times I_1$ | 62.9 d | 26.4 bc | 3.25 d | 15.6 ab |
| $S_4 \times D_1 \times I_1$ | 61.9 e | 27.0 b | 3.31 cd | 15.2 bc |
| $S_1 \times D_2 \times I_1$ | 65.2 b | 24.9 d | 2.94 ef | 15.5 ab |
| $S_2 \times D_2 \times I_1$ | 64.5 bc | 26.0 c | 3.24 d | 15.4 b |
| $S_3 \times D_2 \times I_1$ | 63.0 d | 26.8 b | 3.53 c | 15.3 bc |
| $S_4 \times D_2 \times I_1$ | 63.0 d | 27.6 ab | 3.74 b | 15.0 c |
| $S_1 \times D_3 \times I_1$ | 65.5 b | 25.5 c | 3.29 cd | 15.4 b |
| $S_2 \times D_3 \times I_1$ | 64.8 bc | 26.3 bc | 3.57 bc | 15.1 bc |
| $S_3 \times D_3 \times I_1$ | 64.5 bc | 27.0 b | 3.69 bc | 14.9 c |
| $S_4 \times D_3 \times I_1$ | 64.1 c | 27.8 a | 3.94 ab | 14.7 c |
| $S_1 \times D_1 \times I_2$ | 67.8 a | 23.5 e | 2.96 ef | 15.8 ab |
| $S_2 \times D_1 \times I_2$ | 63.5 cd | 24.0 e | 3.17 d | 15.6 ab |
| $S_3 \times D_1 \times I_2$ | 62.9 d | 25.4 cd | 3.54 c | 15.5 ab |
| $S_4 \times D_1 \times I_2$ | 62.3 dd | 25.7 c | 3.95 ab | 15.4 b |
| $S_1 \times D_2 \times I_2$ | 65.7 b | 24.2 de | 3.06 e | 15.7 ab |
| $S_2 \times D_2 \times I_2$ | 65.0 b | 24.4 de | 3.35 cd | 15.5 ab |
| $S_3 \times D_2 \times I_2$ | 64.6 bc | 24.7 d | 3.73 b | 15.3 bc |
| $S_4 \times D_2 \times I_2$ | 63.4 cd | 25.3 cd | 3.99 ab | 15.1 bc |
| $S_1 \times D_3 \times I_2$ | 65.0 b | 26.1 bc | 3.14 de | 15.6 ab |
| $S_2 \times D_3 \times I_2$ | 64.2 bc | 26.7 b | 3.52 c | 15.4 b |
| $S_3 \times D_3 \times I_2$ | 63.8 c | 27.5 ab | 3.78 b | 14.9 c |
| $S_4 \times D_3 \times I_2$ | 62.6 d | 28.0 a | 4.12 a | 14.7 c |

* There is no difference in the significance level of 0.01 between the averages shown with the same letter in each column. SC: Starch Content (%), ZSV: Zeleny Sedimentation Value (mL), ADF: Acid Detergent Insoluble Fiber (%), and NDF: Neutral Detergent Insoluble Fiber (%).

### 3.2. Starch Ratio and Protein Content

For the starch ratios, the water table depths and groundwater salinity levels were found to be highly significant ($p \leq 0.01$). The effects of the irrigation treatments on the starch ratios were not found to be significant (Table 3). In terms of the water table depths, the greatest starch content (64.3%) was obtained from the $D_3$ and $D_2$ levels and the lowest (63.3%) from the $D_1$ level (Table 4). Groundwater salinity negatively influenced the starch ratios. The greatest starch content (65.1%) was obtained from the $S_1$ level and the lowest (62.8%) from the $S_4$ level.

The protein contents of wheat were considerably affected by the water table depths and salinity (Table 3). In terms of the water table depths, the protein ratios varied between 12.5% ($D_1$) and 13.2% ($D_3$) (Table 4). Decreasing protein ratios were observed with the rising water table levels. In terms of the groundwater salinity levels, the greatest protein ratio was obtained from the $S_4$ (13.4%) salinity level and the lowest from the $S_1$ (12.3%) salinity level. The effects of the irrigation treatments on the protein ratios were not found to be significant. In terms of the D × I interactions, the greatest protein ratio (13.1%) was obtained from the $D_3$ x $I_1$ combination and the lowest from the $D_1 \times I_2$ combination (Table 5).

### 3.3. Zeleny Sedimentation Value, Wet Gluten, and Fat Content

Significant differences were observed in the Zeleny sedimentation values at the different irrigation, water table depth, and groundwater salinity treatments ($p \leq 0.01$) (Table 3). The sedimentation value was measured as 26.2 mL in the irrigated treatments ($I_1$) and as 25.4 mL in the non-irrigated treatments ($I_2$) (Table 4). In terms of the water table depths, the greatest sedimentation value (26.9 mL) was obtained from the $D_3$ treatment, respectively, followed by the $D_2$ (25.5 mL) and $D_1$ (25.2 mL) treatments. In terms of the groundwater

salinity, the greatest sedimentation value (26.9 mL) was obtained from the $S_4$ salinity level and the lowest (24.8 mL) from the $S_1$ salinity level. Increasing sedimentation values were observed with the increasing groundwater salinity levels. In terms of the S × D interactions, the greatest sedimentation value (27.9 mL) was obtained from the $S_4 \times D_3$ combination and the lowest (23.9 mL) from the $S_1 \times D_1$ combination (Table 5). In terms of the D × I interactions, the greatest sedimentation value was obtained from the $D_3 \times I_1$ combination. The greatest sedimentation value was obtained from the $S_4 \times D_3 \times I_2$ combination (Table 6).

For the wet gluten contents, the experimental treatments were significant ($p \leq 0.01$) (Table 3). In terms of the salinity levels, the greatest wet gluten content (27.6%) was obtained from the $S_4$ level, respectively, followed by the $S_3$ (26.9%), $S_2$ (26.2%), and $S_1$ (25.8%) salinity levels (Table 4). Increasing gluten contents were observed with the increasing salinity levels. The wet gluten contents decreased with the decreasing water tables (respectively, $D_3$, $D_2$, and $D_1$). The wet gluten content was measured as 26.8% in the irrigated treatments ($I_1$) and 26.4% in the non-irrigated treatments ($I_2$). In terms of the salinity × irrigation interactions, the greatest wet gluten content (28.1%) was obtained from the $S_4 \times I_1$ combination and the lowest (25.7%) from the $S_1 \times I_2$ combination (Table 5).

The water table depth and salinity had significant effects on the fat content, while the effects of the irrigation treatments on the fat content were not significant. Salinity decreased the fat content by 1.22%, 0.61%, and 3.05% in the S2, S3, and S4 treatments, respectively, compared to the $S_1$ treatment. The highest plant fat content value (1.67) was determined in $D_2$, while the lowest (1.56 cm) was observed from the $D_1$ treatment.

### 3.4. Ash Content, Acid Detergent Fiber, and Neutral Detergent Fiber

There were significant differences in the ash contents at the different irrigation, water table depth, and salinity treatments (Table 3). The ash contents increased with the increasing water table depths ($D_1$, $D_2$, and $D_3$, respectively, with 1.57, 1.60, and 1.65%) (Table 4). The ash contents also increased with the increasing salinity levels ($S_1$, $S_2$, $S_3$, and $S_4$, respectively, with 1.49, 1.57, 1.65, and 1.73%). The ash content was measured as 1.62% in the irrigated plots ($I_1$) and as 1.58 in the non-irrigated plots ($I_2$) (Table 5).

There were highly significant differences in the acid detergent fiber (ADF) and neutral detergent fiber (NDF) values of the experimental treatments ($p \leq 0.01$) (Table 3). In terms of the salinity levels, the greatest ADF and NDF (1.73 and 3.85%, respectively) values were obtained from the $S_4$ salinity level. In terms of the water table depth, the lowest ADF and NDF (1.57 and 3.27%, respectively) values were obtained from the $D_1$ level (Table 4).

### 3.5. Biplot Analysis

A biplot analysis allows researchers to visually assess the relationships between the experimental treatments and investigated parameters and offers some advantages over the correlation analysis revealing relationships only between two traits [26]. The classification of the investigated traits based on the experimental treatments and changes in the investigated parameters of the experimental treatments are presented in Figure 2. In the biplot analysis, two principal components explained 77.0% of the total variation (PC1 57.1% and PC2 19.9%) (Figure 1). As shown in the biplot graph, the hectoliter weight, wet gluten, sedimentation, and protein ratio were positioned in the upper-right section of the graph. Since the vector angles of these traits were less than 90°, there were significant positive relationships between these parameters. There were significant positive relationships between the NDF and starch ratios. The starch content, NDF values, $S_1$ and $S_2$ salinity levels, and $D_1$ and $D_2$ water table levels were placed in the upper-left section of the graph. Therefore, it was thought that the $S_1$, $S_2$, $D_1$, and $D_2$ treatments were prominent for starch and NDF. There was a significant positive relationship between the ash content and ADF. The ash content, ADF, $S_3$ and $S_4$ salinity levels, and $D_3$ water table level were placed into the lower-right section of the biplot graph. According to this result, there was a significant positive relationship between the ash content and ADF. The fat content was placed into the

lower-left section of the graph. Approaching the center of the graph, the $D_2$, $S_2$, $I_1$, and $I_2$ treatments were prominent for more than one trait (Figure 2).

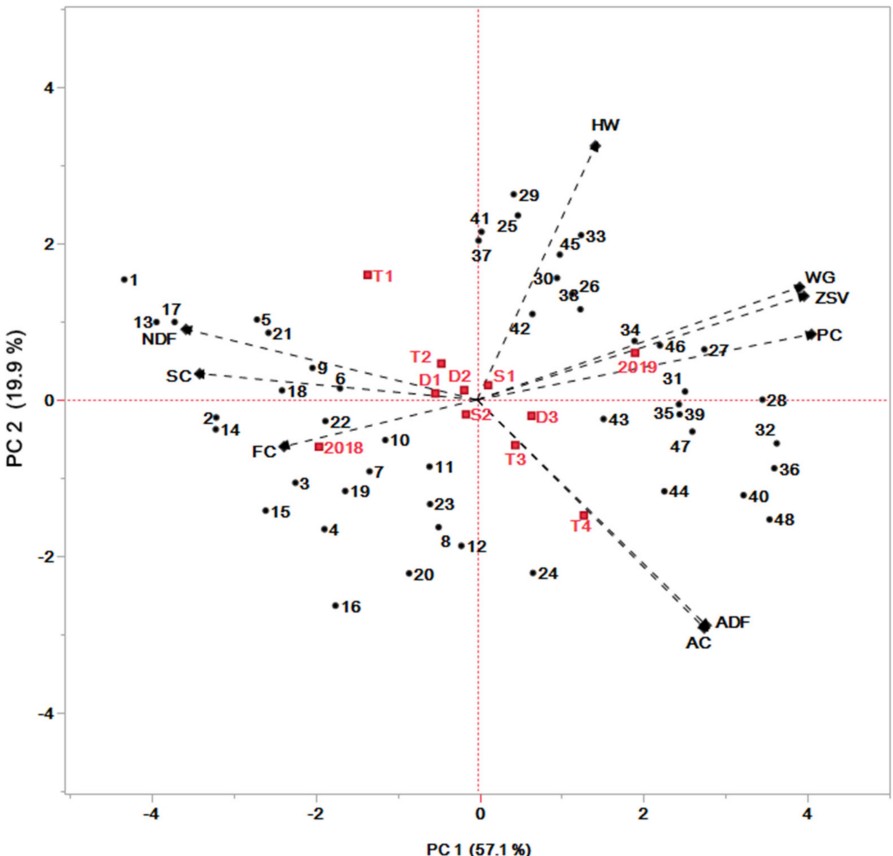

**Figure 2.** Categorization of the examined features by the biplot analysis method and the relationships of the traits examined.

### 3.6. Correlation Analysis

The correlations between the investigated parameters are depicted in Table 7. The hectoliter weight had significant positive correlations with the protein ratio, Zeleny sedimentation and wet gluten content. Additionally, the protein ratio was positively correlated with the Zeleny sedimentation, wet gluten, ash content, and ADF values while negatively associated with the fat content, starch ratio, and NDF values. The wet gluten content had significant positive correlations with the ash ratio and ADF values and significant negative correlations with the oil ratio, starch ratio, and NDF value.

**Table 7.** Correlation coefficients between the features and significance levels*.

|  | HW | PC | ZSV | WGC | ASH | FC | SC | ADF |
|---|---|---|---|---|---|---|---|---|
| **PC** | 0.44 ** | | | | | | | |
| **ZSV** | 0.50 ** | 0.98 ** | | | | | | |
| **WGC** | 0.52 ** | 0.96 ** | 0.99 ** | | | | | |
| **ASH** | −0.17 | 0.48 ** | 0.38 ** | 0.34 * | | | | |
| **FC** | −0.06 | −0.42 ** | −0.46 ** | −0.45 ** | −0.18 | | | |
| **SC** | 0.01 | −0.65 ** | −0.67 ** | −0.65 ** | −0.49 ** | 0.78 ** | | |
| **ADF** | −0.18 | 0.51 ** | 0.40 ** | 0.39 ** | 0.86 ** | −0.12 | −0.46 ** | |
| **NDF** | −0.27 | −0.72 ** | −0.67 ** | −0.66 ** | −0.69 ** | 0.24 | 0.49 ** | −0.71 ** |

* The $p < 0.05$ and ** $p < 0.01$ levels are significant. HW = Hectoliter Weight (kg), PC = Protein Content (%), ZSV =Zeleny Sedimentation Value (mL), WGC = Wet Gluten Content (%), ASH = Ash (%), FC = Fat Content (%), SC = Starch Content (%), ADF = Acid Detergent Insoluble Fiber (%), and NDF = Neutral Detergent Insoluble Fiber (%).

## 4. Discussion

The present study found that the hectoliter weights were affected by the abiotic stress factors and decreased with the increasing stress conditions. Salinity, drought, waterlogging, and other abiotic stresses adversely affect the biochemical and physiological processes in plants and cause deterioration of the grain quality [21]. The change in the grain quality varies according to the amount of stress. As a result of the accumulation of salts in the root zone of the plant, the water and mineral intakes of the plants decrease, and this causes the grain quality to deteriorate. The hectoliter weight is an important physical quality criterion designating, especially, the flour yield. The hectoliter weights were negatively influenced as the water table depths approached the soil surface and decreased with the increasing groundwater salinity levels. Wheat plants had greater hectoliter weights under irrigated conditions than under non-irrigated conditions. It was reported in previous studies that hectoliter weights were mostly influenced by cultural practices and biotic and abiotic stressors [13,27–29]. The hectoliter weights of bread wheat cultivars change between 77.90 and 79.86 kg [30].

The starch and protein contents have significant effects on the bread quality. In our study, the starch content values of the grain quality were considerably affected by the groundwater depths and salinities. Previous researchers indicated that the starch contents greatly varied with the growing conditions [13,31]. The starch contents were increased with the increasing water table depths (1.0, 1.4, 1.8, 2.2, 2.6, and 3.0 m), and also, researchers reported that the starch contents varied between 70.2 and 77.8% in the first year and between 71.2 and 77.2% in the second year [32]. Especially in bread wheat, the protein quantity and quality are among the most important quality traits [14,33]. The protein content of bread wheat should be $\geq$11% [30]. Differences in the grain protein contents were mostly attributed to climate factors and cultural practices [34–36]. The protein ratios decrease with the increasing abiotic salt concentrations [37–39]. It was reported that the protein ratios increased with the increasing water table depths [32] and salinity [40].

The bread volume increases with the increasing sedimentation values; thus, bread wheat is desired to have high sedimentation values [41]. Different groundwater depths and salinities under with or without irrigation conditions considerably influenced the sedimentation values. According to this result, the sedimentation values were increased with the increasing groundwater depths and salinities. Additionally, the $I_1$ (with irrigation) conditions had a higher sedimentation value in comparison to the $I_2$ conditions. This situation could be attributed to the salt accumulation and soil moisture variations in the root zone according to the different groundwater depths, salinities, and irrigation conditions. Previous researchers also indicated that the sedimentation values varied with the environmental conditions and cultivars [13,35].

Wet gluten is the most important quality characteristic of bread wheat and refers to the bread quality of wheat. [13]. In our study, the gluten content values increased with the increasing salinity and depth levels of the groundwater. Additionally, the influence of the $I_1$ condition was greater than the $I_2$ condition on wet gluten. The gluten protein gives rising and elasticity attributes to wheat flour [42]. Increased gluten contents were reported with the increasing irrigation water and soil salinity levels [43–45]. The ash content in wheat is closely related to the flour yield, and the climatic factors can change the ash content. Increasing abiotic stress conditions such as salinity and drought increase the ash contents [46]. The ash contents of wheat species may vary with the climate and soil conditions [16,42].

Significant variations were reported in the quality traits of wheat farming practiced under abiotic stress factors induced especially by high temperatures and insufficient precipitation [13]. There were positive correlations between the ash content and ADF values and highly significant negative correlations between the starch ratio and NDF values. Previous researchers indicated that the ADF and NDF values of bread wheat genotypes generally varied with the genotypes and environmental conditions [28,47]. There were highly significant positive correlations between the fat ratio and starch ratio. Similar correlations were

also reported in previous studies [7,10,29,30]. There were significant negative correlations between the ADF and NDF values.

## 5. Conclusions

The initiation of irrigations in bread wheat farming fields, unconscious irrigations, and excessive fertilizations generated both drainage and salinity problems in these fields. Such problems reduced not only the yields but, also, some quality attributes of bread wheat. In the present study, the water table depth and groundwater salinity levels had highly significant correlations with the irrigation treatments. The investigated quality parameters were positively influenced by increasing the water table depths from 30 cm to 80 cm. The greatest values for the quality traits were obtained from the 80-cm (D3) water table depth treatments. With the increasing groundwater salinity levels from the $S_1$ to $S_4$ levels, the NDF, hectoliter weight, fat ratio, and starch ratios decreased and the protein ratio, sedimentation value, wet gluten content, ash ratio, and ADF values increased. In bread wheat, the protein ratio, sedimentation, and wet gluten contents are important quality traits for the milling industry. Increasing the salinity levels positively influenced these traits. Groundwater salinity may increase the accumulation of salts in the soil and can generate persistent damages in the soil structure. However, the present findings revealed the better quality of bread wheat cultivated in saline lands. Bread wheat cultivation is recommended to be done in places with deep water table levels and irrigation opportunities.

**Author Contributions:** Conceptualization, H.A. (Hasan Akay), H.A. (Hakan Arslan), Z.M. and İ.S.; methodology, H.A. (Hasan Akay), H.A. (Hakan Arslan), Z.M. and İ.S.; software, M.S.K.; validation, H.A. (Hasan Akay), H.A. (Hakan Arslan), Z.M. and İ.S.; formal analysis, H.A. (Hasan Akay); investigation, Ö.D.E.K.; resources, E.Ö., Ö.D.E.K. and M.S.K.; data curation, H.A. (Hasan Akay), H.A. (Hakan Arslan), Z.M., Ö.D.E.K. and İ.S.; writing—original draft preparation, İ.S., H.A. (Hasan Akay), Z.M., H.A. (Hakan Arslan), Ö.D.E.K., E.Ö. and M.S.K.; writing—review and editing, İ.S., H.A. (Hasan Akay), Z.M., H.A. (Hakan Arslan), Ö.D.E.K., E.Ö. and M.S.K.; visualization, M.S.K. and H.A. (Hasan Akay); supervision, H.A. (Hasan Akay), H.A. (Hakan Arslan), Z.M. and İ.S.; project administration, H.A. (Hakan Arslan) and İ.S.; funding acquisition, H.A. (Hasan Akay) All authors have read and agreed to the published version of the manuscript.

**Funding:** This study was supported by the Scientific and Technological Research Council of Turkey (TÜBİTAK), project number 116O492.

**Institutional Review Board Statement:** Not applicable.

**Informed Consent Statement:** Not applicable.

**Data Availability Statement:** The data presented in this study are available on request from the corresponding author.

**Conflicts of Interest:** The authors declare no conflict of interest.

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
