# Peer review of "Effects of Different Water Table Depth and Salinity Levels on Quality Traits of Bread Wheat"

_agriculture, doi:10.3390/agriculture11100969_

Round 1

Reviewer 1 Report

The manuscript requires substantial revision in the description of materials and methods, in the presentation of the tables and in the discussion of the results. There are several changes required

Reviewer 2 Report

Sezer et al. conducted a study on the effect of water depth and salinity levels on the quality of the wheat crop. Scientists raised a very topical topic related to the changing climate. Despite this, the obtained results do not add much more than what is known to farmers. I suggest the authors improve the Results and Discussion chapter with a more elaborate discussion of the results obtained. In addition, the title requires a format change, the name of the country should be added in the authors' affiliations, and the description of Table 3 should be more detailed. Extra spaces appear in the text in many places.

Reviewer 3 Report

This study examines the alterations of different water depths and salinity on wheat quality. The paper could provoke interest in the researchers in the field of plant stress physiology and agriculture, however, the results and discussion section is confusing and unclear.

Major comments and observations:

The results of the article should be described in the order of the table, and the results of different treatments should be described in different paragraphs.

The manuscript is proposed to be revised as follows:

  1. Results and discussion section should be described separately.
  2. The result section needs to be given a clear subtitle.

Round 2

Reviewer 1 Report

still are some  methodological error that need to be improved

99-102: is not clear the classification of the soil.

99-102 Please refer the methods for soil analysis

 99-102: KG da-1 what measure is?

L 120-122: is the same sentences of L 170-173. I any case, the bock design ned t be better explained, especially for how statistical analysis are treated (the description refer a split split plot, the analysis refer a randomised block a three ways.

L 155: N Units

L 156-158: considering that diammonium phosphate is supported, that support Phosphorus and nitrogen, Urea complete the N nutrition. Please clarify

L 163: grain quality

L 164-168: Unit of measures

Tab 3: According the number of lysimeter, the repetition are 3, but the DF are 4 (that mean that the repetitions are 5). Please check.

Tab. 4 – Unit of measure. And the mean of the years?

Tab. 5: and the years interactions

Tab. 6: as Tab. 5

Tab 4, 5, 6 the parameter are different, please check

212-216: please check (NC?). put the legend in same order of the columns?

285: space between references

Tab 7: SV or ZSV?

The paper need a deep revision. Please verify accuracy the manuscript

Reviewer 3 Report

This study explored the effects of different salt concentrations and water depths on wheat quality. After the author’s manuscript has been revised, the introduction section has been supplemented to elaborate the relationship between salt concentration and drought on wheat quality; the title given in the results and discussion section can clearly explain the correlation between the test parameters of wheat quality and the treatments.  The manuscript was revised to make the results and discussion more complete.
